# Impact of Protein-Enriched Plant Food Items on the Bioaccessibility and Cellular Uptake of Carotenoids

**DOI:** 10.3390/antiox10071005

**Published:** 2021-06-23

**Authors:** Mohammed Iddir, Juan Felipe Porras Yaruro, Emmanuelle Cocco, Emilie M. Hardy, Brice M. R. Appenzeller, Cédric Guignard, Yvan Larondelle, Torsten Bohn

**Affiliations:** 1Department of Population Health, Luxembourg Institute of Health, 1445 Strassen, Luxembourg; mohammed.iddir@lih.lu (M.I.); juanfelipe.porras@etu.emse.fr (J.F.P.Y.); emilie.hardy@lih.lu (E.M.H.); Brice.Appenzeller@lih.lu (B.M.R.A.); 2Louvain Institute of Biomolecular Science and Technology, UCLouvain, 1348 Louvain-la-Neuve, Belgium; yvan.larondelle@uclouvain.be; 3École Nationale Supérieure des Mines, 42023 Saint-Étienne, France; 4Environmental Research & Innovation Department, Luxembourg Institute of Science and Technology, L-4422 Belvaux, Luxembourg; emmanuelle.cocco@list.lu (E.C.); cedric.guignard@list.lu (C.G.)

**Keywords:** INFOGEST protocol, tetraterpenoids, micellization, small intestinal uptake, emulsions, lipid droplets, mixed micelles, free fatty acids, bioavailability

## Abstract

Carotenoids are lipophilic pigments which have been associated with a number of health benefits, partly related to antioxidant effects. However, due to their poor solubility during digestion, carotenoid bioavailability is low and variable. In this study, we investigated the effect of frequently consumed proteins on carotenoid bioaccessibility and cellular uptake. Whey protein isolate (WPI), soy protein isolate (SPI), sodium caseinate (SC), gelatin (GEL), turkey and cod, equivalent to 0/10/25/50% of the recommended dietary allowance (RDA, approx. 60g/d), were co-digested gastro-intestinally with carotenoid-rich food matrices (tomato and carrot juice, spinach), and digesta further studied in Caco-2 cell models. Lipid digestion, surface tension and microscopic visualization were also carried out. Co-digested proteins positively influenced the micellization of carotenes (up to 3-fold, depending on type and concentration), especially in the presence of SPI (*p* < 0.001). An increased cellular uptake was observed for xanthophylls/carotenes (up to 12/33%, *p* < 0.001), which was stronger for matrices with an initially poor carotenoid micellization (i.e., tomato juice, *p* < 0.001), similar to what was encountered for bioaccessibility. Turkey and cod had a weaker impact. Significant interactions between carotenoids, lipids and proteins were observed during digestion. Co-digested proteins generally improved lipid digestion in all matrices (*p* < 0.001), especially for carrot juice, though slight decreases were observed for GEL. Protein impact on the surface tension was limited. In conclusion, proteins generally improved both carotenoid bioaccessibility and cellular uptake, depending on the matrices and carotenoid-type (i.e., carotene vs. xanthophylls), which may be relevant under specific circumstances, such as intake of carotenoid-rich food items low in lipids.

## 1. Introduction

Carotenoids belong to the category of tetraterpenoids. They can be divided into two main classes; carotenes, which are of linear carbon structure, and xanthophylls, carrying at least one oxygen molecule in their structure. These phytochemicals are naturally synthetized by plants, algae, bacteria and fungi [1,2]. In addition to exhibiting a color spectrum from yellow to red, carotenoids are essential pigments in photosynthetic organisms, as they absorb violet and blue-green light. They also act as a photoprotector against overexposure to light, as well as contributing to attracting animals for pollination and dispersal of plant seeds [3,4].

However, humans and other animals do not synthetize carotenoids de novo, i.e., their presence in these organisms is exclusively due to dietary intake. The physiological functions of these secondary plant compounds in the human body appear to be complex and are the topic of much debate. Numerous studies have shown that carotenoids exert protective and/or preventive roles in human health and diseases, which have in part been related to their antioxidant functions, quenching reactive oxygen species such as singlet oxygen or lipid-peroxides [5]. Carotenoid plasma concentrations were associated with reduced risk of chronic diseases such as cardiovascular diseases [6] and type 2 diabetes [7], as well as certain types of cancer [8]. In addition, the presence of certain types of carotenoids in the eye is associated with reduced risk of developing several ocular diseases such as age-related macular degeneration and cataracts, caused by photo-damage of the retina, especially among the elderly [9,10]. Furthermore, certain types of carotenoids, i.e., β-carotene, α-carotene, and β-cryptoxanthin are cleaved into retinal by β-carotene oxygenase 1 (BCO1) and/or β-carotene oxygenase 2 (BCO2), which could place carotenoids as the main source of vitamin A for vegetarians/vegans and in many countries with a low consumption of animal-based food items [11]. Some of these metabolites could also interact with nuclear and transcription factors such as NF-κB and Nrf-2, with anti-inflammatory and anti-oxidant effects, respectively [12].

The close relation between carotenoid consumption and clinical outcomes has recently accelerated the pace of carotenoid research directed towards human nutrition, including questions related to bioavailability. In fact, carotenoids go through several stages following their dietary intake, starting by the release from the food matrix to allow their dissolution in lipid droplets in the stomach [13,14] and their later incorporation, following processing by gastric-and pancreatic lipase activity [15] into mixed micelles in the small intestine [16]. This is largely thought to determine the fraction of carotenoids that are available for absorption, which takes place via either passive diffusion and/or via transporter-dependent processes [17], and is a good approximation of their bioavailability [18], a prerequisite for further physiological functions.

However, the bioavailability of carotenoids is highly variable [19,20], and is known to be influenced by several intrinsic and extrinsic factors [21], including the food matrix. For instance, several studies have concluded that dietary lipids [22,23], divalent minerals [24,25] and dietary fiber [26] could influence carotenoid bioavailability. However, the potential influence of co-consumed dietary proteins on digestion processes of carotenoids has started to gain attention more recently. It is acknowledged that, during gastro-intestinal (GI) digestion, proteins go through structural modifications, which maximizes interactions between hydrophobic and hydrophilic phases, which may aid in emulsifying apolar dietary constituents [27]. These structural modifications have been observed for example in soy protein isolate (SPI), generating short protein chains, resulting in a better amphiphilic property and a decrease in the interfacial tension and the size of the emulsion droplets [28]. Another study reported that protein segments generated by the digestion of sodium caseinate (SC) formed a stable emulsion, carrying lipophilic compounds such as β-carotene, preventing its oxidation and facilitating its transfer into mixed micelles, which increased the bioaccessibility of β-carotene [29]. However, while proteins have received much interest from pharmaceutical and food industries, especially for encapsulation and delivery of lipophilic compounds [30,31], including carotenoids [32], very little is known about the influence that proteins could have on the bioavailability of carotenoids, under conditions resembling the in vivo environment.

Results from our previous investigations demonstrated that whey protein isolate (WPI) was able to modulate β-carotene bioaccessibility during in vitro GI digestion, improving its bioaccessibility when the amount of co-digested dietary lipids is limited. However, a negative effect was observed under insufficient digestion parameters (i.e., limited concentrations of digestive enzymes and bile), likely via interactions during the processing of lipid droplets into mixed micelles by limiting the extent of lipolysis, therefore hindering carotenoid transfer [33]. Similarly, a study by Goupy et al. showed that the transfer of pure carotenoids to oil droplets was dramatically impaired when the emulsions were stabilized by bovine serum albumin [34]. In addition, we have reported in a recent study that various proteins (WPI, SPI, SC and gelatin (GEL)) had both positive and negative effects on the bioaccessibility of isolated carotenoids (β-carotene, lycopene and lutein) [35]. These influences depended not only on the type of carotenoids, where higher polarity was reflected in a higher degree of micellization, but also on the type and concentration of proteins where their digestibility and hydrophobicity were related to their emulsifying properties.

In the present study, we aimed to investigate the effect of several, frequently consumed proteins with different structure, and thus digestibility, on the bioaccessibility and cellular uptake of carotenoids from selected food sources. Toward this end, pure proteins (WPI, SPI, SC and GEL) or protein-rich food matrices (turkey and cod), at different concentrations (equivalent to the addition of 0/10/25/50% of the recommended dietary allowance (RDA)) were co-digested with food matrices rich in carotenoids (tomato juice, spinach and carrot juice), using the INFOGEST consensus digestion model [36]. Following simulated digestion, we examined the impact of proteins on the cellular uptake of carotenoids, by employing Caco-2 cells as a model for the intestinal epithelium. For obtaining additional mechanistic insights, parameters such as surface tension, free fatty acid release and microscopic visualization of the emulsions were also studied.

## 2. Materials and Methods

### 2.1. Enzymes, Chemicals and Carotenoid Standards

Digestive enzymes such as pepsin from porcine gastric mucosa (powder, ≥250 U/mg, measured as trichloroacetic acid-soluble products using hemoglobin as substrate, Art. No. P7000), pancreatin from porcine pancreas (activity equivalent to 4x USP specifications. Art. No. P1750), as well as porcine bile extract (Art. No. B8631) were ordered from Sigma-Aldrich (Overijse, Belgium).

All chemicals were of analytical grade or superior. Potassium chloride, potassium phosphate, sodium bicarbonate, sodium chloride, magnesium chloride hexahydrate, ammonium carbonate, sodium hydroxide solution (1 M), calcium chloride dihydrate, ammonium acetate, butylated hydroxytoluene (BHT), Nile red, fluorescein isothiocyanate isomer I (FITC) and phenolphthalein were acquired from Sigma-Aldrich, while hydrochloric acid solution (1 M), and acetic acid were obtained from VWR (Leuven, Belgium). Acetonitrile, methanol, ammonium acetate and dichloromethane were purchased from Biosolve (Dieuze, France). Methanol (MeOH), acetonitrile (ACN), dichloromethane (DCM) and methyl tert-butyl ether (MTBE) were purchased from Carl Roth GmbH+ Co. KG (Rotisol^®^, Karlsruhe, Germany). The solvents hexane, acetone and methanol were obtained from VWR, while diethyl ether was from Sigma-Aldrich. In addition, 18 MΩ water was used throughout the study, it was either prepared with a purification system from Millipore (Brussels, Belgium) or procured from B. Braun Medical SA (Sempach, Switzerland).

Standards of *all-trans-*β-carotene (β-Car, powder, ≥97%), *all-trans*-lycopene (Lyc, powder, ≥85%), α-carotene (α-Car, powder, ≥97%), neoxanthin (Neo, powder, ≥97%) and *trans*-β-Apo-8′-carotenal (IS, powder, ≥96%) were purchased from Sigma-Aldrich; lutein (Lut, powder, ≥95%), zeaxanthin (Zea, powder, ≥98), and β-cryptoxanthin (β-Cry, powder, ≥97%) were obtained from Extrasynthese (Genay, France); violaxanthin (Vio, powder, ≥95%), (9*Z*)-β-carotene ((*9Z*)-β-Car, powder, ≥95%), phytoene (PTE, oily, ≥95%), and phytofluene (PTF, oily, ≥95%) were acquired from CaroteNature GmbH (Ostermundigen, Switzerland).

The penicillin/streptomycin mixture (Pen/Strep, Lonza, Basel, Switzerland), trypsin/EDTA solution (Lonza), Dulbecco’s modified Eagle’s medium (DMEM + GlutaMAX^TM^, Gibco^TM^) and heat-inactivated fetal bovine serum (FBS, Gibco) were acquired from Fisher Scientific (Illkirch, France). Non-essential amino acids (100× NEAA, Lonza) and Dulbecco’s phosphate buffered saline (DPBS, Lonza) were purchased from VWR.

### 2.2. Food Matrices

Whey protein isolate (WPI) was acquired from Pure Nutrition USA (95% purity, Oxnard, CA, USA), and soy protein isolate (SPI) was obtained from Self Omninutrition^®^ (≥90% purity, Stockholm, Sweden), while casein sodium salt (SC) from bovine milk (≥85% purity) and gelatin (GEL) from bovine skin (≥70% purity) were both purchased from Sigma-Aldrich. All protein solutions were prepared by dissolving 0, 3, 7.5 and 15 g/L of the final digestion volume, corresponding to approx. 0, 10, 25 and 50% of the recommended dietary allowance (RDA, approx. 60 g/d for human adults), respectively [35]. Cod fillet (average nutritional values per 100 g: 0.7 g fat/0 g carbohydrates:/17.8 g proteins/0.1 g salt) and sliced turkey fillet (average nutritional values: 1.2 g fat/<0.5 g carbohydrates:/25 g proteins/0.13 g salt) were purchased from a local supermarket (Cora, Bertrange, Luxembourg). Pieces of fillet were cut and chopped as much as possible using a kitchen blender (Moulinette chopper, Moulinex, Ecully, France). Aliquots containing 2.19 g of cod fillet or 1.56 g of turkey fillet on average were prepared, representing the equivalent of 50% RDA for total proteins (when digesting this in 25 mL and extrapolating this to 2 L of digestive fluid in the GI tract). Aliquots were placed in thermo-resistant glassware containing boiling water (≥75 °C) until cooked (approximately 15 min.), and, after being cooled down, they were flushed with argon and stored at −80 °C until in vitro GI digestion.

Organic 100% carrot juice (Delhaize, other ingredients: 1% concentrate from lemon juice), 100% tomato juice (Delhaize, other ingredients: 0.5% salt), and frozen spinach (Iglo, other ingredients: 0.04% salt) were purchased from a local supermarket (Delhaize, Strassen, Luxembourg). Using a kitchen blender, the spinach was vigorously grinded. All tested matrices were aliquoted into portions of approx. 35 g in 50 mL Falcon tubes, flushed with argon and stored at −80 °C until further analysis.

### 2.3. Simulation of Gastro-Intestinal Digestion of Carotenoids

In vitro simulated digestion was carried out according to the harmonized INFOGEST protocol [36], with some modification [33,35]. As the matrices were either liquid or ground, and were low in complex carbohydrates, the oral phase was omitted. The gastric digestion procedure was carried out following the recommended ratio of matrix (protein solution + food matrix) to simulated gastric fluids of 50:50 (*v*/*v*). Each sample consisted of a freshly prepared pure protein solution with the desired concentration equivalent to 0, 10, 25 or 50% of the RDA for proteins, or previously thawed cod and turkey fillet aliquots at a concentration equivalent to 50% of the RDA for proteins, plus 4 g of plant food matrix (tomato juice, spinach or carrot juice). The in vitro gastric passage was performed as reported earlier with a final volume of 13 mL [35]. Similarly, the simulated intestinal digestion was carried out by maintaining the ratio of chyme to simulated intestinal fluids of 50:50 (*v*/*v*). Again, similarly, the remaining steps of the intestinal digestion were performed in the same manner as previously described, with a final GI digestion volume of 26 mL [35]. At the end of the simulated digestion, aliquots of 12 mL were removed from the digesta and centrifuged at 3300× *g* for 1 h (4 °C). Then, 5 mL were collected from the middle aqueous phase and filtered through a 0.2 um nylon membrane syringe filter. The filtered micellar fractions present in the aqueous phase were taken for the extraction of carotenoids [24], while aliquots of complete digesta samples were flushed with argon and stored at –80 °C for further analysis.

### 2.4. Cell Culture and Uptake Experiments

Parental Caco-2 cells, originating from a human colorectal carcinoma [37], were obtained from LGC Standards (Molsheim, France, Art. No. ATCC-HTB-37). Caco-2 cells used in this study were between passages 66 and 75. The cell culture and uptake procedures were adapted from earlier studies [38,39] and are briefly described below.

#### 2.4.1. Cell Culture

Caco-2 cells were grown in 75 cm^2^ flasks (Nunclon^TM^ Delta Surface, Roskilde, Denmark) at 37 °C in an atmosphere of 5% CO_2_ (Thermo Electron Corporation, Waltham, MA, USA), in DMEM+GlutaMAX^TM^ supplemented with 10% FBS, 1% of NE AA and 1% of Pen/Strep (10,000 U/mL of each antibiotic). The medium was changed regularly until the flasks reached approximately 80% confluence [38]. Thereafter, cells were seeded in six well-plates (Nunclon^TM^) at a density of 5 × 10^4^ cells/cm^2^ for 15 days to allow a proper cell differentiation [38], similar as described in earlier trials where differentiation was followed visually and based on TEER experiments [40,41].

#### 2.4.2. Cellular Uptake Experiment

Medium was removed and cells were washed with PBS, and 3 mL of medium diluted complete digesta (obtained at the end of GI digestion, centrifuged for 2 min at 1000× *g* to remove larger particles) (digesta/DMEM; 1/8; *v*/*v*) were added before incubation for 4 h. The digesta was removed after incubation, and cells were washed with cold PBS and detached with medium as described earlier [39]. Cellular uptake experiments were performed in triplicates, which were pooled in a 15 mL Falcon tube before further extraction of carotenoids.

### 2.5. Extraction of Carotenoids

The extraction procedure from plant food matrices was carried out from an aliquot of 4 g of each matrix, using successively hexane/acetone mixture (2/1, *v*/*v*), hexane and diethyl ether [42]. For the green leafy matrix, i.e., spinach, the extraction was preceded by a step of saponification with 1 mL of 30% aqueous KOH [42]. Similarly, the filtered micellar fraction was extracted using hexane/acetone mixture (2/1, *v*/*v*), and then repeated twice with only pure hexane [35].

Regarding the extraction of carotenoids from Caco-2 cells, the fresh pools of three replicates were centrifuged, and the cells were resuspended in ice cold water, followed by a brief vortex and sonication in the dark [38]. Multiple extractions with a hexane/acetone mixture (2/1, *v*/*v*), hexane and diethyl ether were performed, all followed by a quick vortex and 2 min of sonication before centrifugation for 2 min. at 4000× *g* (4 °C). An aliquot of all combined extracts was collected and dried under a stream of nitrogen (TurboVap LV from Biotage^®^, Uppsala, Sweden). The dried extracts were flushed with argon and stored at –80 °C until analysis.

### 2.6. Quantification of Carotenoids

The percentage of carotenoid micellization was used as a measure of bioaccessibility. Both the bioaccessibility and cellular uptake were expressed as the percentage of carotenoid concentration measured in the dried extracts following in vitro digestion and cellular uptake experiments, respectively, compared to the initial concentration measured in the original matrix.

#### 2.6.1. Spectrophotometric Measurements

As a first evaluation of the impact of the proteins studied (i.e., cod, GEL, SC, SPI, turkey, and WPI, at a concentration of 50% RDA) on carotenoid micellization, the total carotenoid concentration of the bioaccessible fraction was calculated by a method based on the mean absorption coefficients in hexane and mean absorption wavelength, and concentration was calculated as described by Biehler et al. [43].

#### 2.6.2. HPLC Analysis

For the analysis of carotenoid extracts from the bioaccessible fraction obtained following in vitro GI digestion of carotenoid-rich matrices in the absence or presence of selected protein types (GEL, SC, SPI, and WPI), the dried extracts were re-dissolved in a total volume of 500 μL of MTBE:MeOH (30/70, *v*/*v*). In order to enhance dissolution of more apolar carotenoids, first, 150 μL of cold MTBE were added to the extracts, and then 350 μL of MeOH, both steps were followed by a brief vortex and sonication. The Internal Standard (IS, *trans*-β-Apo-8′-carotenal) was added to each sample with a final concentration of 200 ng/mL. The same procedure was applied to the dried extracts from food matrices, which were re-dissolved in a total volume of 6 mL of MTBE:MeOH (30/70, *v*/*v*). An aliquot of 80 µL was transferred into an HPLC amber vial (equipped with a glass insert). HPLC separation of carotenoids was carried out using an Agilent 1260 Infinity U-HPLC instrument (Agilent Technologies, Santa Clara, CA, USA), equipped with a C30 reversed phase column (2.6 μm particle size, 100 mm length, 2.1 mm diameter, Thermo Fisher Scientific, Breda, The Netherlands). The mobile phase consisted of water/MeOH (60/40, *v*/*v*) with 30 mM of ammonium acetate as eluent A and ACN:DCM (85/15, *v*/*v*) as eluent B. Elution gradient was as follows: 0 min. 42% B; 4 min. 48% B; 5 min. 52% B; 11 min. 52% B; 13 min. 75% B; 18 min. 90% B; 40 min. 90% B; 41 min. 42% B. The flow rate was fixed at 0.8 mL/min., the injection volume was 10 μL, and column temperature was 28 °C. Seven point-linear calibration curves were prepared with external standards for each compound, with concentrations ranging from 60 to 500 ng/mL. Peaks were integrated at 286 nm (PTE), 350 nm (PTF), 440 nm (NEO and VIO), 450 nm (ZEA, LUT, α-CAR, β-CAR, and (*9Z*)-β-CAR), 455 nm (β-CRY and IS) and at 470 nm (LYC). Carotenoids were detected with a diode array detector, and identified by comparing each carotenoid’s retention time and absorption maxima [44], with those of the available standards.

#### 2.6.3. LC-MS Analysis

For the analysis of dry carotenoids residues obtained following cellular uptake experiments, the cellular extracts were reconstituted in 1 mL of MeOH/DCM (60/40, *v*/*v*), vortexed thoroughly, and an aliquot of 50 µL was transferred into an amber vial (equipped with a glass insert) containing 5 ng of IS. Each sample of cellular extract was prepared and analysed in triplicate (*n* = 3). Working solutions with concentration levels ranging from 0.01 µg/L to 1 mg/L were prepared. In order to prepare matrix-matched calibration curves, twelve aliquots of untreated cellular extracts were resuspended in a MeOH/DCM mixture (as for the samples). The injection volume was 2 µL, and carotenoid concentrations in the cellular extracts were determined according to external calibration curves. The analysis were performed with a Waters Acquity UPLC H-Class PLUS equipped with a BEH C_18_ column (100 mm, 2.1 mm ID, 1.7 µm particle size, Waters, Milford, MA, USA) combined with a Waters Xevo TQ-S tandem mass spectrometer operating with electrospray ionization (Appendix A). Similar to that carried out by Bukowski et al. [45], the mobiles phases were: MeOH/water (90/10, *v*/*v*) + 20 mM ammonium acetate (mobile phase A) and ACN/DCM/MeOH (70/20/10, *v/v/v*) + 20 mM ammonium acetate + 0.3% acetic acid (mobile phase B). The column temperature and sample compartment were 28 °C and 20 °C, respectively. The flow rate applied was set at 0.35 mL/min. The solvent gradient was as follows: 1.6 min. of initial composition at 100% A, 1 min. of linear increase to 100% B, composition hold from 2.6 to 9 min. and 7 min. back to initial composition for next injection and equilibration. The parameters set for MS/MS acquisition were the following: capillary voltage at 2 kV, source and desolvation temperatures at 150 °C and 650 °C, respectively, cone gas flow at 150 L/h, desolvation gas flow at 1200 L/h, collision gas flow at 0.15 mL/min and nebulizer set at 6 bar.

### 2.7. Free Fatty Acid Determination, Surface Tension and Confocal Laser Scanning Microscopy

Complete digesta aliquots stored at −80 °C after GI digestion of spinach and carrot juice were used to evaluate lipid hydrolysis by measuring the amount of free fatty acid (FFA) release. This was determined by Cayman’s Free Fatty Acid Fluorometric Assay (Cayman Chemical, Art. No. 700310, Ann Arbor, MI, USA) according to the manufacturer’s protocol.

Surface tension of digesta samples were determined by the weight-drop method as previously described [46].

Confocal imaging of emulsion structures was carried out after the gastric phase and at the end of the GI digestion, using the confocal laser scanning microscope (Zeiss LSM 880, Airyscam SR, Jena, Germany). Nile red and FITC were mixed into the aliquots of digesta for fat and protein staining, respectively, while carotenoids were visualized by their natural fluorescence. An argon 488 nm laser excited the fluorescent dyes while a 403 nm laser excited the carotenoids (β-carotene used as reference). The emitted light of the samples was collected as following: 450–490 for carotenoids, 500–540 nm for proteins, and 590–650 nm for lipids. The resultant images were processed as described previously [35].

### 2.8. Statistical Analyses and Data Treatment

Unless otherwise specified, all data are presented as mean ± SD. Linear mixed models were created to study the effects of added proteins or proteins from food matrices on the bioaccessibility and cellular uptake of individual carotenoids. For this purpose, non-normally distributed data (following Q-Q plots and box plots) or data with unequal variances were log-transformed. Linear mixed models thus contained the amount of protein (0, 10, 25 or 50% RDA), type of protein (WP, SP, GEL, SC, turkey and cod), type of matrix (tomato juice, carrot juice, spinach), free fatty acid release and surface tension as well as the type of carotenoid as fixed factors, while bioaccessibility or cellular uptake were the observed dependent factors. *p*-values below 0.05 (2-sided) were considered significant. Following Fisher-F-tests, all group wise comparisons were carried out by Fisher-protected LSD tests (three groups) or Tukey’s (>3 groups). Following significant interactions, models were re-run keeping one of the factors in the interaction term constant in order to re-do all group-wise comparisons. Therefore, findings for pooled results (several conditions combined) are generally reported first, followed by individual comparisons. All analyses were carried out in SPSS (vs. 25, IBM statistics, Chicago, IL, USA).

## 3. Results

### 3.1. Carotenoid Patterns for Each Food Matrix

Spinach contained the highest amount of total quantified carotenoids (15.4 mg/100 g), followed by tomato juice (10.9 mg/100 g) and carrot juice (7.6 mg/100 g, Table 1). While carotenes (α-Car, β-Car, and Lyc) were the major carotenoids in tomato and carrot juices, representing 74% and 81% of total carotenoid content, respectively, xanthophylls (Neo, Vio, Zea, Lut and β-Cry) represented approx. 74% of total quantified carotenoids in spinach. In juice matrices, the contribution of the colorless carotenoid phytoene and phytofluene to the total detected carotenoid content was approx. 26% in tomato juice, and approx. 18% in carrot juice, while their contribution was rather negligible in spinach (<1% of total carotenoid content) (Table 1).

### 3.2. Influence of Various Proteins on the Bioaccessibility of Total Carotenoids

Under control conditions (no added proteins), the overall bioaccessibility of carotenoids (all carotenoids combined) differed significantly between the food matrices (*p* < 0.001, Appendix A). Following in vitro GI digestion, overall carotenoid micellization was higher from spinach (19.0 ± 0.4%) than from carrot juice (12.1 ± 0.3%) and tomato juice (2.3 ± 0.3%).

Following linear mixed model analysis, the effect of proteins on the bioaccessibility of total carotenoids was statistically significant (*p* < 0.001), with highest carotenoid micellization in the presence of SC and GEL, followed by turkey, SPI and cod, and finally WPI.

Their addition during digestion at various concentrations had a significant influence compared to the control condition (*p* < 0.001, except at 10% RDA concentration). When investigating all proteins together, a higher bioaccessibility was found at a concentration of 25% RDA, while a slight but significant decrease of total carotenoid micellization was obtained following the addition of a concentration of 50% RDA. Regarding the matrices, the proteins increased carotenoid micellization in tomato juice (from 2.3 to 3.4%), while a slight decrease was observed in carrot juice (from 12.1 to 9.3%), and in spinach (from 19.0 to 16.9%).

More specifically, only co-digested turkey proteins improved the bioaccessibility of carotenoids from spinach compared to the control (*p* < 0.05) in general, while other protein types significantly decreased it (*p* < 0.001, Figure 1). The opposite effect was found for digested tomato juice, i.e., turkey proteins had a negative effect on the bioaccessibility, and, except for the neutral effect of cod proteins, all other types of proteins significantly improved overall carotenoid bioaccessibility compared to the control (*p* < 0.001, Figure 1). In carrot juice, a significant decrease was found in the presence of co-digested cod proteins, WPI, SPI (*p* < 0.001), and SC (*p* < 0.05, Figure 1), while turkey proteins and GEL had no effect.

Thus, the proteins seemed to improve the bioaccessibility of those matrices with an initially poor carotenoid micellization, such as tomato juice. In contrast, matrices with an originally rather higher carotenoid bioaccessibility, such as spinach and carrot juice, showed a more pronounced decrease in bioaccessibility when co-digested with proteins.

### 3.3. Influence of Proteins on the Bioaccessibility of Individual Carotenoids

#### 3.3.1. General Effects on Different Carotenoids

In the absence of proteins, the bioaccessibility of individual carotenoids in spinach followed the order: lutein+zeaxanthin > neoxanthin > β-carotene > violaxanthin, while their contributions to total bioaccessible carotenoids followed a similar order (Table 2). For carotenoid species present in tomato juice, their bioaccessibility followed the order: lutein+zeaxanthin > β-carotene > phytoene > phytofluene > lycopene (Table 2). However, the contribution of each individual carotenoid in the micellar fraction differed compared to their micellization. In carrot juice, the bioaccessibility of individual carotenoids followed the order: lutein+zeaxanthin > phytoene > phytofluene > α-carotene > β-carotene (Table 2).

The effect of carotenoid species was statistically significant (*p* < 0.001), similar to that of the matrix (*p* < 0.001). In addition, a significant interaction between matrix type and carotenoid type was encountered (*p* < 0.001), thus the effects were further studied at individual carotenoids and food items.

#### 3.3.2. Comparison across Matrices—Overall Effects

To better compare the effects of proteins on carotenoid bioaccessibility across all tested matrices, β-carotene as well as lutein+zeaxanthin were chosen, as they were the only carotenoids present in all of the investigated matrices (Table 2). Likewise, further focus was placed on pure proteins regarding studying their influence on the bioaccessibility of β-carotene and lutein+zeaxanthin, as they showed a stronger impact compared to cod and turkey proteins on the bioaccessibility of total carotenoids.

In addition to the significant interaction between the food matrix and carotenoid species, the effects of protein type and concentration were also found to be significant (*p* < 0.001), and the interaction between all these effects was statistically significant (*p* < 0.001). Thus, the effects on the bioaccessibility of lutein+zeaxanthin and β-carotene were further studied per protein and matrix. Of note, when studying all matrices combined, the micellization of lutein+zeaxanthin in the presence of proteins was higher than that of β-carotene (*p* < 0.001).

#### 3.3.3. Comparison across Matrices—Lutein+Zeaxanthin

The bioaccessibility of lutein+zeaxanthin differed significantly between the plant food matrices (*p* < 0.001), and was found to be higher from tomato juice than from carrot juice and from spinach (lowest micellization, Appendix A). The presence of proteins had a negative effect on the micellization of lutein+zeaxanthin in all investigated matrices (Figure 2). In the presence of SC (considering all protein concentrations combined), the bioaccessibility of lutein+zeaxanthin was higher compared to WPI (*p* < 0.001), with no significant effect with GEL, while lowest micellization of lutein+zeaxanthin was observed in the presence of SPI (*p* < 0.001). Additionally, the co-digested proteins negatively influenced the bioaccessibility in a concentration-dependent manner (*p* < 0.001).

The co-digestion of WPI together with either spinach or tomato juice had no influence on lutein+zeaxanthin bioaccessibility, except for a slight decrease in spinach at 50% RDA (*p* < 0.05). However, its presence during the digestion of carrot juice reduced, in a concentration-dependent manner, the micellization of lutein+zeaxanthin (*p* < 0.05, Figure 2). Regarding the effect of SPI, a drastic decrease of lutein+zeaxanthin bioaccessibility in a concentration-dependent manner in all tested matrices was found (*p* < 0.001). The co-digestion of SC to the digestion of spinach resulted in a slight significant decrease of lutein+zeaxanthin micellization, while it had no influence when added to carrot juice or tomato juice (except for a slight increase at 50% RDA for tomato juice, *p* < 0.05). Regarding GEL, its presence had no influence on the bioaccessibility of lutein+zeaxanthin from tomato juice, while a significant decrease was found at higher concentrations (25 and 50% RDA) in spinach and carrot juice (*p* < 0.001).

#### 3.3.4. Comparison across Matrices—β-Carotene

β-Carotene bioaccessibility also differed significantly between the investigated matrices (*p* < 0.001). Similar as for lutein+zeaxanthin, the bioaccessibility of β-carotene was higher from tomato juice compared to carrot juice, while being lowest for digested spinach (Appendix A).

The effect of protein type significantly influenced β-carotene recovery in the micellar fraction (*p* < 0.001). The presence of SPI (considering all concentrations combined) resulted in highest β-carotene bioaccessibility, followed by SC, WPI and finally GEL. While at 10% RDA proteins had no influence on β-carotene bioaccessibility, the addition of proteins at higher concentrations (25 and 50% RDA) resulted in an increased β-carotene micellization, compared to the control condition (*p* < 0.001). Regarding the different matrices, the presence of proteins strongly improved β-carotene micellization in tomato juice, as well as in spinach, while the micellization of β-carotene from carrot juice was most negatively affected (Figure 2).

The co-digestion of WPI with either spinach or tomato juice significantly increased β-carotene bioaccessibility (*p* < 0.001), while, in carrot juice, the micellization significantly dropped (*p* < 0.05, Figure 2). Similarly, the presence of SPI during digestion positively affected the bioaccessibility of β-carotene from spinach and tomato juice (*p* < 0.001), while a slight decrease was found for carrot juice (*p* < 0.001). Adding SC significantly deceased β-carotene micellization in spinach, while a positive effect was found in tomato and carrot juice (*p* < 0.001), except at the highest concentration, where a slight decrease was found for carrot juice (*p* < 0.001). Regarding GEL, a negative effect on the bioaccessibility of β-carotene for spinach and carrot juice was observed. However, a concentration-depending increase was found when co-digested with tomato juice (*p* < 0.001, Figure 2).

Overall, changes regarding bioaccessibility were somewhat limited, especially for the xanthophylls (small relative decreases of ca. 1/10th when considering all matrices, proteins and their concentrations combined), while up to almost a three-fold increase was seen under certain conditions for carotenes.

### 3.4. Influence of Various Proteins on Cellular Uptake of Total Carotenoids

Under control conditions, fractional cellular uptake of total carotenoids (the sum of the individual carotenoids, compared to the original amount present in the matrix) differed significantly between the investigated matrices (*p* < 0.001, Appendix A). Overall, carotenoid uptake was higher from spinach (1.3% ± 0.08) than from tomato juice (0.7% ± 0.03) and carrot juice (0.2% ± 0.01).

The effect of proteins on the cellular uptake of total carotenoids (considering different concentrations combined) was statistically significant (*p* < 0.001), with no significant difference between WPI, SPI and GEL, though SC resulted in higher total carotenoid uptake compared to other protein types (*p* < 0.001). The addition of various proteins and their concentrations resulted in an increased total carotenoid uptake, compared to the control condition (no added protein, *p* < 0.001), following the order 10 > 50 > 25% RDA. Similarly, cellular uptake of total carotenoids was significantly different between the investigated matrices in the presence of proteins (*p* < 0.001). Co-digested proteins appeared to influence tomato juice more strongly, with an increase from 0.7 to 1%, while a slight decrease of total carotenoid uptake from carrot juice and spinach was found (Appendix A).

In spinach (considering all protein concentrations combined), co-digestion with GEL resulted in the highest cellular uptake of total carotenoids, followed by WPI and SC and SPI (*p* < 0.001). Similarly, in the digested carrot juice, the presence of GEL and SC resulted in a higher overall carotenoid uptake than SPI (*p* < 0.05), while WPI significantly reduced carotenoid uptake compared to other tested proteins (*p* < 0.05). In tomato juice, a significant difference was found between the tested proteins (*p* < 0.001), with higher uptake of total carotenoids in the presence of WPI, followed by SPI, GEL and SC.

Though cellular uptake was low overall, and effects of protein additions apparently limited, relative increases of up to almost two-fold were seen under certain conditions, though also decreases to about half of the original levels. However, the effect of protein on the uptake of total carotenoids was comparable to the bioaccessibility, with an overall positive effect on carotenoids in tomato juice, while a negative influence was observed for especially carrot juice, and also for spinach.

### 3.5. Influence of Proteins on Cellular Uptake of Individual Carotenoids

#### 3.5.1. General Effects on Different Carotenoids

Under control conditions, fractional cellular uptake of individual carotenoids in spinach followed the order: lutein+zeaxanthin > β-carotene > neoxanthin + violaxanthin > β-cryptoxanthin, while their contributions to total amounts taken up differed (Table 3). For carotenoids present in tomato juice, their fractional cellular uptake followed the order: lutein+zeaxanthin > β-carotene > α-carotene > phytoene > phytofluene > lycopene. (Table 3). In carrot juice, the cellular uptake (%) of individual carotenoids followed the order: lutein+zeaxanthin > phytofluene > β-carotene > α-carotene > lycopene (Table 3).

The effect of matrix as well as carotenoid species were statistically significant (*p* < 0.001), thus the effects were studied further at individual carotenoid species and food items.

#### 3.5.2. Comparison across Matrices—Overall Effects

Again, lutein+zeaxanthin as well as β-carotene were selected to better compare the effects of proteins at various concentrations on carotenoid cellular uptake across all tested matrices (Table 3).

The interaction between protein type, protein concentration, carotenoid species and matrix type was statistically significant (*p* < 0.001), similar as reported for the bioaccessibility, thus the effects on the uptake of lutein+zeaxanthin and β-carotene were further studied for individual matrices and proteins. The cellular uptake of lutein+zeaxanthin in the presence of proteins was higher than that of β-carotene (*p* < 0.001, all matrices and protein concentrations considered).

#### 3.5.3. Comparison across Matrices—Lutein+Zeaxanthin

Under control conditions, a higher uptake of lutein+zeaxanthin was obtained from tomato juice, followed by carrot juice, and spinach, with a significant difference between all matrices (*p* < 0.001, Table 3). A slight difference was found between protein types regarding their influence on the cellular uptake of lutein+zeaxanthin, which followed the order: GEL > WPI > SPI > SC. In addition, 10 and 25% RDA protein addition significantly increased the uptake of lutein+zeaxanthin, up to 10.4%, compared to the control (7.7%, *p* < 0.001), while 50% RDA slightly decreased lutein+zeaxanthin uptake (*p* < 0.05).

Contrary to what was observed for bioaccessibility, the presence of proteins had a positive effect on the uptake of lutein+zeaxanthin in all investigated matrices, with the strongest increase in tomato juice (up to 22.0% cellular uptake), compared to the control (17.1%, Figure 3). Positive effects were more limited for carrot juice and lowest for spinach, in line with bioaccessibility showing least negative effects for tomato juice and strongest reductions for spinach.

#### 3.5.4. Comparison across Matrices—β-Carotene

Cellular uptake of β-carotene differed between the tested matrices (*p* < 0.001), with highest uptake from tomato juice, followed by spinach and carrot juice (Table 3). The investigated proteins influenced differently cellular uptake of β-carotene (*p* < 0.001, except in the presence of WPI and GEL), with a higher uptake observed in the presence of SPI, and followed by GEL, WPI and SC, when considering all protein concentrations combined. Their presence at various concentrations enhanced β-carotene uptake in a concentration-dependent manner, with an increase up to 6.0%, compared to the control (2.7%). The influence of proteins on the cellular uptake appeared to mimic the observed impact on the bioaccessibility of β-carotene, i.e., being positive in all investigated matrices, with the strongest increase in tomato juice (from 7.6 to ca. 12.5%, Figure 3).

Overall, co-digested proteins had a positive influence on cellular uptake, with a relative increase of up to 12% for xanthophylls, and up to 33% for carotenes (all matrices, proteins and concentrations considered).

### 3.6. Lipid Analysis

Under control conditions, the release of FFAs differed depending on the type of the matrix (*p* < 0.001), and was higher in spinach (0.019 µM), compared to carrot juice (0.013 µM).

Adding proteins at various concentration to digestion differentially influenced FFA release (*p* < 0.001, Figure 4). Co-digestion of proteins with the investigated food matrices influenced differently lipid digestibility compared to controls (*p* < 0.001), with higher FFA release in the presence of SC, followed by WPI and SPI, and finally GEL (both matrices combined). Additionally, their presence during the digestion influenced FFA release in a concentration-depended manner (*p* < 0.001), with a relative increase of up to 86% compared to the control condition. Similarly, the presence of proteins had a positive effect on lipid digestibility in all investigated matrices, with an average increase of up to 46%.

In spinach, a better lipid digestibility was obtained in the presence of SC, WPI and SPI, compared to GEL which had a negative influence on FFA release (*p* < 0.001). Globally, increasing protein concentrations slightly improved FFA release, with a significant effect at 50% RDA, compared to the control (*p* < 0.001). Co-digestion of WPI with spinach improved significantly lipid digestibility at all tested concentrations, while SC and SPI improved FFA release only at 50% RDA (*p* < 0.001, Figure 4). However, GEL at 10 and 25% RDA significantly lowered the digestion of lipids (*p* < 0.05, Figure 4A).

In carrot juice, however, a higher FFA release was obtained in the presence of SC, as well as by WPI and SPI, while GEL had a negative impact (*p* < 0.001). Similar as found for spinach, increasing the concentration of proteins enhanced the release of FFAs during the digestion of carrot juice, being significant at 25 and 50% RDA (*p* < 0.001). The presence of SC in the digesta enhanced lipid digestibility in a concentration-depended manner (*p* < 0.001), with an almost 3-fold increase. For SPI and WPI, only higher concentrations significantly improved the release of FFAs (*p* < 0.001). With less impact, the addition of GEL to the carrot juice digesta lowered the release of FFAs at a concentration of 10% RDA, while there was no significant influence at higher concentrations (Figure 4B).

Thus, proteins appeared to influence more strongly carrot juice, although slight decreases were observed in the presence of GEL.

### 3.7. Physicochemical Characteristics of the Digesta

#### 3.7.1. Microscopic Visualization

Microscopic visualization also revealed the extent of lipid digestibility through the confocal imaging of emulsion structures taken after gastric and GI digestion (Figure 5). The presence of proteins in the simulated digestion resulted in different emulsion structures, though all proteins remarkably appeared to reduce the size of the emulsified particles after complete GI digestion, compared to emulsions obtained in digesta collected after the gastric phase.

At concentrations of 25 and 50% RDA, WPI and SPI showed a higher degree of aggregation compared to SC and GEL. This difference was even accentuated at the end of the gastric stage. A co-localization of the labelled compounds was observed in all digesta, indicating an interaction between carotenoids, lipids and proteins during the in vitro GI digestion (Figure 5).

#### 3.7.2. Surface Tension

Although the overall effect of co-digested proteins was low and limited (an average decrease from 46.9 to 45.0 dyn/cm, Figure 6), the effect of added proteins apparently had more impact on the surface tension of digested spinach than liquid matrices (up to 8% relative decrease).

## 4. Discussion

In this study, we tested whether various frequently consumed types of proteins could interact with carotenoid bioaccessibility or cellular uptake. Indeed, especially discriminative interactions between xanthophylls and carotenes were found, with up to three-fold improvement in carotene bioaccessibility but likewise also reductions of over 50% in xanthophyll bioaccessibility. Such potential changes of bioavailability could be important toward assuring optimal intake of carotenoids, especially in sight of their disease-related aspects including acting as direct or indirect antioxidants.

WPI, SPI, SC and GEL were co-digested with carotenoid-rich food matrices (tomato juice, carrot juice and spinach), following the INFOGEST model for static in vitro GI digestion [36,47]. When upscaled to a daily food intake, amounts were approx. equivalent to 0, 10, 25 and 50% of the recommended dietary allowance (RDA) of protein intake of adults. The food matrices represented regularly consumed fruits and vegetables, providing a wide range of carotenoids. It was found that the bioaccessibility of total carotenoids was highest from spinach, while being lowest for tomato juice. The investigated matrices differed in carotenoid content, i.e., carotenes represented the major quantified carotenoids in tomato and carrot juice (74 and 81%, respectively), while xanthophylls represented approx. 74% of total carotenoids in spinach. As carotenoid bioaccessibility has been inversely correlated with their hydrophobicity [48], the findings of highest bioaccessibility of polar xanthophylls (mostly lutein+zeaxanthin) from spinach vs. the nonpolar carotenes in carrot juice (mostly β-carotene) and tomato juice (mostly lycopene) are not too surprising.

However, in addition to an effect on carotenoids, matrix effects were also registered, as lutein+zeaxanthin and β-carotene from liquid matrices represented better bioaccessibility compared to their homologues in spinach, following the sequence tomato juice > carrot juice > spinach. The physical state of the matrix in which carotenoids are incorporated likely affects aspect of release during digestion and hence their bioaccessibility [49]. Fiber content in spinach is known to limit carotenoid micellization [26], and carotenoids in spinach have been reported to be mostly present within chloroplasts [50], while carotenoids occur in a solid crystalline deposition form in both tomato and carrot chromoplasts [51]. Though the original carotenoid location has likely changed due to the more drastic heat treatment, some cells may remain intact even after juice processing [52]. It has been shown that carrots have small cell sizes paired with thicker, very fibrous, and denser cell walls, compared to the very large cells and thin cell walls of tomato [53], which could be one of the reasons why tomato juice exhibited a higher total bioaccessibility than carrot juice, similar to that observed earlier [54].

Regarding the influence of proteins, it has been acknowledged that proteins are predominantly adsorbed at the oil/water (o/w) interfaces via their hydrophobic segments, which are dissolved in the oil phase [55]. Surrounding the lipid droplets, proteins aid in emulsifying liposoluble dietary constituents [56] and enable the incorporation of non-polar components into emulsions during digestion [57]. In the present study, the presence of proteins improved total carotenoid bioaccessibility of tomato juice, while decreasing that of spinach and carrot juice. We have previously reported these two diverging effects of proteins on the bioaccessibility of pure carotenoids during in vitro GI digestion [35]. While co-digested proteins enhanced the bioaccessibility of β-carotene by up to 50% compared to the protein-free control, it hampered lutein bioaccessibility. As the digested matrices differed greatly in their content and profile of carotenoids, this likely arbitrated their distribution in the lipid droplets, and thus the extent of micellization [58]. Borel et al. have reported that polar xanthophylls such as zeaxanthin were preferentially solubilized at the lipid droplet surface, while the apolar β-carotene was solubilized almost exclusively in the core of the lipid droplets [13]. This may lead to a negative interaction of the adsorbed proteins and xanthophylls at the o/w interface, especially in view of proteins often rearranging their molecular conformation to facilitate adsorption of hydrophobic groups at the interface in order to obtain an energetically favorably conformation [59]. Furthermore, only a limited amount of carotenoids can be solubilized within mixed micelles, due to the limited loading capacity [60]. This may explain, at least in part, why the presence of proteins during digestion could interfere with the trend of bioaccessibility, being inversely correlated with their aqueous solubility [48]. Therefore, it is proposed that, on one hand, the high original bioaccessibility of xanthophylls was hampered by the presence of adsorbed proteins, limiting their incorporation into mixed micelles, while, on the other hand, proteins, via their emulsifying properties, enhanced the initially poorly micellized carotenes by increasing their solubilisation and transition into the mixed micelles.

The co-digested proteins influenced differentially the bioaccessibility of carotenoids. All investigated proteins showed their ability to enhance lipid digestibility, while their emulsifying capability was somehow limited as illustrated by small changes in surface tension. Though all proteins appeared to reduce the size of the emulsified particles after complete GI digestion, the presence of proteins in the simulated digestion resulted in different emulsion structures. WPI and SPI showed a higher degree of aggregation compared to SC and GEL. In our previous work, we reported that the major protein fraction of WPI (i.e., β-lactoglobulin) remained intact only at the early stage of the intestinal digestion, while SPI digestibility was incomplete due to low-size polypeptide fragments remaining even following complete GI digestion [35]. The ensuing undigested peptides could have resulted in binding of carotenoids.

Indeed, the hydrophobicity of the investigated proteins may play a decisive role in the extent of carotenoid micellization. It has been reported that the solubility of SPI is low, [52] leading to high surface hydrophobicity [61]. The reason has been attributed to the denaturation induced by heating, drying, and storage conditions [62,63]. A recent study showed that β-carotene bioaccessibility was highly related to the type of proteins used to stabilize the emulsions [64]. SPI-based emulsions possessed the highest bioaccessibility compared to WPI and SC, which were insufficient to stabilize the emulsions, leading to coalescence. Similarly, SPI was successfully used as a nano-carrier to enhance the stability and bioaccessibility of β-carotene in a concentration-dependent manner, through hydrophobic intermolecular interactions (i.e., van der Waals forces and hydrogen bonds) occurring mainly in the hydrophobic sites of SPI [65,66]. The high surface hydrophobic nature and aggregated state were suggested to impart SPI a key role to perform as nano-carriers for lipophilic compounds, including β-carotene. Once denatured by proteases during digestion, the structure of SPI will unfold and the hydrophobic clusters initially buried within the structure of proteins will be exposed, which may foster β-carotene-protein interactions. These findings are in line with our results, as highest β-carotene bioaccessibility was obtained when co-digested with SPI (up to 50% RDA), compared to other proteins, improving micellization by almost three-fold (e.g., in spinach).

Regarding the bioaccessibility of lutein+zeaxanthin, in addition to their negative interactions that may occur at the interface with the adsorbed proteins, the more polar xanthophylls may also show low binding affinity to the hydrophobic sites of proteins, perhaps explaining the strong negative influence of proteins on lutein+zeaxanthin bioaccessibility, with co-digested proteins decreasing their micellization in a concentration-dependent manner (*p* < 0.001). Lowest micellization was observed in the presence of SPI. Conversely, the higher solubility of SC and WPI (also considering the complete hydrolysis of other protein fractions such as α-lactalbumin) may produce more peptides that have amphiphilic structure, and these may be more beneficial for the more polar carotenoids, facilitating their interactions with the micelles [67] and enhancing their bioaccessibility, in line with results from the present study.

Despite the standardized protein amounts, the effect of protein-rich food matrices (i.e., turkey and cod) on the bioaccessibility of total carotenoids was smaller than pure proteins. A slightly higher bioaccessibility was obtained in the presence of turkey, compared to cod. It has been reported that isolated proteins have weaker network structures and more cleavage sites, and are hydrolyzed faster and to a greater extent than protein-rich foods [68]. In addition, co-digested plant food matrices may have an additional impact on the protein network structure of the digested animal matrices, possibly by preventing enzymes from accessing proteins, and therefore reducing protein digestibility [68]. Additionally, a higher protein digestibility was found in turkey-rich meals (60% turkey and 40% poultry products) compared to fish meal (sardine), using casein as the reference protein [69]. Thus, we can only speculate that the limited effect of rich-protein matrices on the bioaccessibility of carotenoids is probably due to the less complete or slowed-down protein digestion.

In line with bioaccessibility findings, fractional cellular uptake of total carotenoids from spinach was higher than from carrot and tomato juices. Also in this case, xanthophylls were taken up better compared to carotenes, reflecting bioaccessibility. A recent study evaluating the bioaccessibility and cellular uptake by *Caco-2* cells of carotenoids from orange peels reported the same findings [70], i.e., the bioaccessibility and cellular uptake of xanthophylls being significantly higher than that of carotenes. Sy et al. indicated that this may be due to the fact that xanthophylls reside at the surface of the mixed micelles, making them potentially more available for cellular uptake [48]. However, when comparing lutein+zeaxanthin and β-carotene fractional cellular uptake across matrices, it was highest from tomato juice and lowest for spinach (except for β-carotene where cellular uptake was lowest from carrot juice). As reported earlier, we hypothesize that these differences are due to differences in the respective carotenoid content between the matrices, and that higher absolute amounts are taken up at a lower percentage, as realized in the present study with the main carotenoid in spinach (β-carotene, cellular uptake 0.13% as opposed from 7.6% from tomato juice), and lutein/zeaxanthin from spinach (cellular uptake 2.5% as opposed to 17.1% in tomato juice), perhaps pointing out the saturation processes including involved transporters for carotenoids, i.e., NPC1L1 for lutein/zeaxanthin and SR-B1 and CD36 for β-carotene [17]. Such transporters could also explain why fractional cellular uptake differed less drastically between lutein+zeaxanthin vs. β-carotene than the bioaccessibility results, a finding that was earlier reported by Kaulmann et al. [38]. In their study, carotenes such as β-carotene indeed showed a better cellular uptake alone compared to xanthophylls such as lutein+zeaxanthin.

Interestingly, somewhat contrarily to bioaccessibility, the presence of proteins had a positive effect on the *Caco-2* cellular uptake of lutein+zeaxanthin (up to 12%), with the strongest increase for tomato juice, while a limited influence for carrot juice followed by spinach was found. The cellular uptake of carotenes appeared to well reflect the impact of proteins on the bioaccessibility of β-carotene, i.e., being positive in all investigated matrices (up to 33%), with the strongest increase in tomato juice, especially in the presence of SPI (50% RDA), with up to a three-fold increase. Thus, despite the negative effect of co-digested proteins on the bioaccessibility of lutein+zeaxanthin, they were better taken up in the presence of proteins. A recent study on astaxanthin-loaded emulsions using WPI as emulsifier indicated that protein-based emulsions had a higher potential for improving the cellular uptake of astaxanthin than free astaxanthin [71]. Similarly, for carotenes, Lu et al. investigated the bioaccessibility and cellular uptake of β-carotene loaded emulsions, and found that emulsifiers such as WPI and SC improved the cellular uptake by facilitating the interaction of micelles with *Caco-2* cells [67].

Regarding the influence of different protein types on the cellular uptake of carotenoids, results are also aligned with bioaccessibility findings, apparently fostering β-carotene–protein interactions and therefore improving its cellular uptake. A higher cellular uptake of β-carotene was obtained in the presence of co-digested SPI, followed by WPI and GEL and finally SC. The high solubility of SC [35] was apparently playing a lesser role than the potentially more hydrophobic SPI digestion products. Contrarily, xanthophylls were better taken up in the presence of more soluble proteins, such as WPI, while the presence of SPI strongly reduced their cellular uptake.

## 5. Conclusions

The addition of proteins to the simulated GI digestion resulted in an improved fractional bioaccessibility in matrices containing higher amounts of nonpolar carotenes, compared to matrices rich in more polar xanthophylls. Proteins with a high surface hydrophobic nature such as SPI may foster β-carotene–protein interactions, and improved micellization close to 3-fold, while negative interactions with xanthophylls at the O/W interface may occur, limiting their incorporation into the mixed micelles. Similarly, co-digested proteins fostered the cellular uptake of carotenes (β-carotene, to up to 33% uptake), and counteracted the somewhat negative effect of proteins on xanthophyll bioaccessibility by improving lutein+zeaxanthin cellular uptake (up to 12%), especially in the presence of soluble WPI and SC, which may produce peptides that have more amphiphilic structure that can be more beneficial for the more polar carotenoids, facilitating the interactions at the surface of the mixed micelles, making them potentially more available for cellular uptake. Regarding proteins from regular food matrices, their effects were more limited regarding altering carotenoid bioaccessibility compared to matrix-free proteins. Additional work on protein-carotenoid interactions is warranted in sight of the potential health-associated benefits of this group of dietary antioxidants.

## Figures and Tables

**Figure 1 antioxidants-10-01005-f001:**
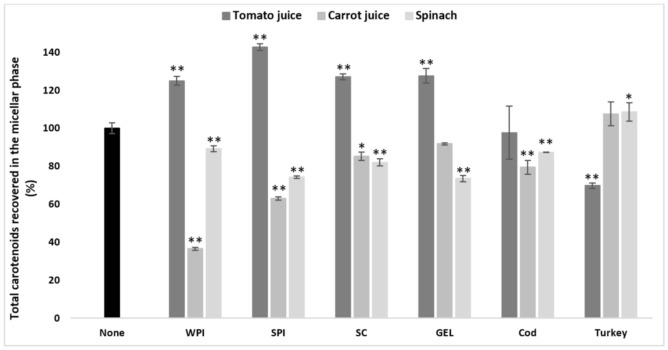
Influence of different proteins on the bioaccessibility of total carotenoids following simulated GI digestions of tomato juice, spinach and carrot juice. Carotenoid rich-food matrices were digested in the absence or presence of different protein rich food items/supplements (cod, turkey, WPI, SPI, SC, and GEL) at a concentration of 50% of protein recommended dietary allowance (RDA, approx. 60 g/d for human adults [35]). Bioaccessibility is expressed as the percentage of carotenoids recovered from the aqueous micellar fraction at the end of the GI digestion, compared to the amount present in each matrix at the beginning of digestion. Bars represent means ± SD of *n* ≥ 4 and *N* ≥ 2 (sets repeated at different days). Columns labelled with either * (*p* < 0.05) or ** (*p* < 0.001) were significantly different from the control condition (no added protein).

**Figure 2 antioxidants-10-01005-f002:**
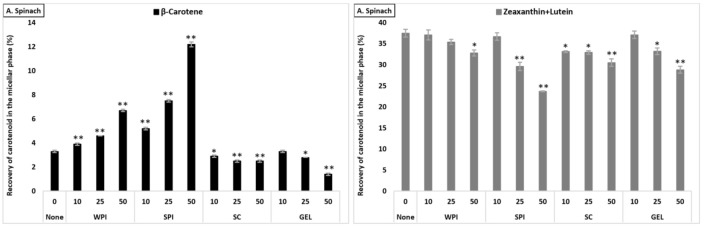
Average bioaccessibility (%) of carotenoids following simulated GI digestions of spinach (**A**), tomato juice (**B**), and carrot juice (**C**). Each bar represents the bioaccessibility of either β-carotene or lutein+zeaxanthin recovered from the aqueous micellar fraction at the end of the in vitro GI digestion. Bioaccessibility is expressed as explained in Figure 1 heading. Bars represent means ± SD of *n* ≥ 4 and N ≥ 2 (sets repeated at different days). Columns labelled with either * (*p* < 0.05) or ** (*p* < 0.001) were significantly different from the control condition (no added protein).

**Figure 3 antioxidants-10-01005-f003:**
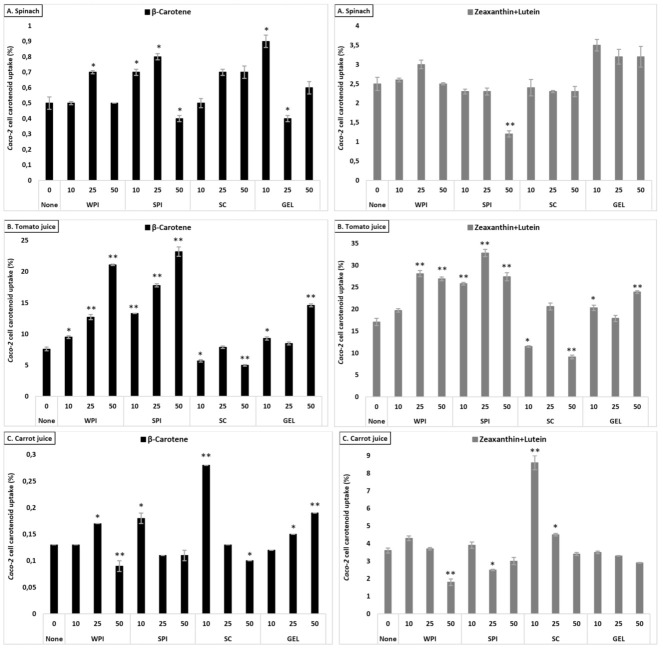
Average cellular uptake (%) of carotenoids from spinach (**A**), tomato juice (**B**), and carrot juice (**C**). The investigated matrices were studied following in vitro GI digestion. Each bar represents the cellular uptake of either β-carotene or lutein+zeaxanthin recovered in the cell fraction compared to the original, matrix content (considering all dilutions). Bars represent means ± SD of *n* = 3. Columns labelled with either * (*p* < 0.05) or ** (*p* < 0.001) were significantly different from the control condition (no added protein).

**Figure 4 antioxidants-10-01005-f004:**
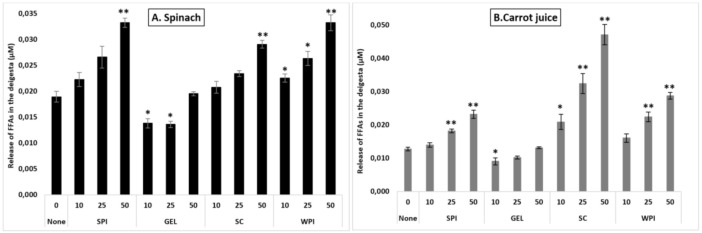
Influence of proteins on the release of free fatty acids (FFAs) during simulated GI digestion of spinach (**A**) and carrot juice (**B**). Carotenoid-rich matrices were co-digested in the presence of WPI, SPI, SC or GEL at various concentrations (0, 10, 25 and 50% of the RDA (approx. 60 g/d for human adults [35])), and the release of FFAs was evaluated at the end of GI digestion (Cayman’s Free Fatty Acid Fluorometric Assay). Values represent means ± SD of *n* = 4. Columns labelled with either * (*p* < 0.05) or ** (*p* < 0.001) were significantly different from the control condition (no added protein).

**Figure 5 antioxidants-10-01005-f005:**
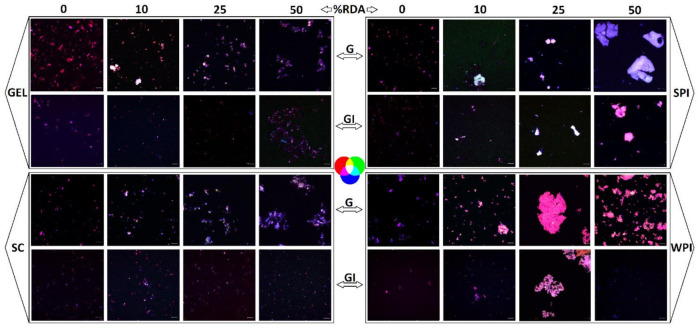
Images taken after gastric (G) and GI digestion with whey protein isolate (WPI), soy protein isolate (SPI), sodium caseinate (SC), and bovine gelatin (GEL). The grey bar equals 10 µm. Confocal imaging of emulsion structures was carried out at room temperature with a confocal laser scanning microscope (Zeiss LSM 880, Airyscam SR, Jena, Germany), using a 63× objective. Fluorescent dyes were excited by an Ar laser (488 nm), and the emitted light was collected at 522 nm for protein and 635 nm for the fat phase. Lipids were labelled with Nile red (red colour) and proteins were labelled with fluorescent isothiocyanate (FITC, green colour), while carotenoids are naturally fluorescent (blue colour).

**Figure 6 antioxidants-10-01005-f006:**
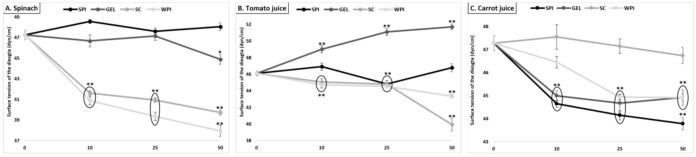
Effect of proteins on the surface tension of the digesta (value of pure water: 71.99 dyn/cm), following GI digestion of spinach (**A**), tomato juice (**B**), and carrot juice (**C**). Carotenoid-rich matrices were digested in the presence of whey protein isolate, soy protein isolate, sodium caseinate or gelatin at various concentrations (0, 10, 25 and 50% of the recommended dietary allowance (RDA, approx. 60 g/d for human adults [35])). The surface tension of digesta samples, pre-conditioned at 25 ± 0.1 °C, was determined by the weight-drop method. Values represent means ± SD of *n* = 4 replicates. Either * (*p* < 0.05) or ** (*p* < 0.001) were significantly different from the con-trol condition (no added protein).

**Table 1 antioxidants-10-01005-t001:** Carotenoid content (mg/100 g) for each tested plant food matrix ^1^, as determined by HPLC.

Food Matrix	Neo	Vio	Lut+Zea	β-Cry	PTF	PTE	α-Car	β-Car	Lyc	Total
Spinach	2.03 ± 0.1	3.29 ± 0.4	6.00 ± 0.5	0.12 ± 0.0	0.02 ± 0.0	0.07 ± 0.0	0.02 ± 0.0	3.67 ± 0.3	n.d.	15.4 ± 0.3
Tomato juice	n.d.	n.d.	0.04 ± 0.0	n.d.	1.15 ± 0.0	1.63 ± 0.1	0.02 ± 0.0	0.25 ± 0.1	7.79 ± 0.4	10.9 ± 0.2
Carrot juice	n.d.	n.d.	0.09 ± 0.0	n.d.	0.88 ± 0.2	0.49 ± 0.1	1.70 ± 0.1	4.40 ± 0.2	0.04 ± 0.0	7.6 ± 0.1

^1^ Each value represents the mean ± SD of *n* ≥ 3 replicates. Neo: neoxanthin; Vio: violaxanthin; Lut + Zea: lutein+zeaxanthin; β-Cry: β-cryptoxanthin; PTF: phytofluene; PTE: phytoene; α-Car: α-carotene; β-Car: β-carotene; Lyc: lycopene. n.d.: not detectable.

**Table 2 antioxidants-10-01005-t002:** Absolute amount ^1^ of carotenoids, bioaccessibility ^2^ as well as their relative contribution to the bioaccessible fraction ^3^ after simulated in vitro gastrointestinal digestion of each tested plant food matrices, as determined by HPLC. Values represent means of *n* ≥ 4 replicates and N ≥ 2 (sets repeated at different days).

Carotenoid	Spinach			Tomato Juice			Carrot Juice		
	Amount mg/100 g	Bioac. %	Contrib. %	Amount mg/100 g	Bioac. %	Contrib. %	Amount mg/100 g	Bioac. %	Contrib. %
Neo	0.4 (±0.0)	22.1 (±0.5)	15.4	n.d.	n.d.	n.d.	n.d.	n.d.	n.d.
Vio	0.1 (±0.0)	2.8 (±0.1)	3.2	n.d.	n.d.	n.d.	n.d.	n.d.	n.d.
Lut + Zea	2.2 (±0.1)	37.5 (±0.9)	77.3	0.03 (±0.0)	66.5 (±1.4)	10.6	0.1 (±0.0)	61.9 (±1.2)	6.1
PTF	n.d.	n.d.	n.d.	0.1 (±0.0)	5.1 (±0.1)	23.0	0.3 (±0.0)	35.6 (±0.8)	34.0
PTE	n.d.	n.d.	n.d.	0.1 (±0.0)	7.6 (±0.1)	48.6	0.3 (±0.0)	53.4 (±1.4)	28.5
α-Car	n.d.	n.d.	n.d.	n.d.	n.d.	n.d.	0.1 (±0.0)	4.8 (±0.1)	8.9
β-Car	0.1 (±0.0)	3.3 (±0.1)	4.1	0.02 (±0.0)	8.0 (±0.2)	8.0	0.2 (±0.0)	4.7 (±0.1)	22.5
Lyc	n.d.	n.d.	n.d.	0.02 (±0.0)	0.3 (±0.0)	9.7	n.d.	n.d.	n.d.
Total car.	2.9 (±0.5)	19.0 (±0.4)	100	0.3 (±0.0)	2.3 (±0.3)	100	0.9 (±0.0)	12.1 (±0.3)	100

^1^ Absolute amount of carotenoids after simulated GI digestion of each tested plant food matrices in the obtained bioaccessible fractions. ^2^ Bioaccessibility expressed as the percentage of carotenoids recovered from the aqueous micellar fraction at the end of the GI digestion, compared to the initial amount present in the undigested test meal. ^3^ Values represent the contribution (%) of each individual carotenoid, relative to the total amount of carotenoids recovered from the aqueous micellar fraction after the in vitro GI digestion of each matrix. Neo: neoxanthin; Vio: violaxanthin; Lut+Zea: lutein+zeaxanthin; PTF: phytofluene; PTE: phytoene; α-Car: α-carotene; β-Car: β-carotene; Lyc: lycopene. n.d.: not detectable.

**Table 3 antioxidants-10-01005-t003:** Absolute amount ^1^ of carotenoids, percentage of their cellular uptake ^2^, as well as their relative contribution to cellular uptake ^3^ to total carotenoids in the fractional cellular uptake phase of carotenoids into Caco-2 cells, as determined by LC-MS-MS. Values represent means of *n* = 3 measurements.

Carotenoid	Spinach			Tomato Juice			Carrot Juice		
	Amount ng/3wells	CellUp. %	Contrib. %	Amount ng/3wells	CellUp. %	Contrib. %	Amount ng/3wells	CellUp. %	Contrib. %
Neo + Vio	14.1 (±0.9)	0.5 (±0.0)	12.4	n.d.	n.d.	n.d.	n.d.	n.d.	n.d.
Lut + Zea	87.3 (±5.6)	2.5 (±0.2)	77.3	4.0 (±0.2)	17.1 (±0.8)	9.8	1.9 (±0.0)	3.6 (±0.1)	28.1
β-Cry	0.1 (±0.0)	0.05 (±0.0)	0.05	n.d.	n.d.	n.d.	n.d.	n.d.	n.d.
PTF	n.d.	n.d.	n.d.	8.1 (±0.5)	1.2 (±0.1)	19.7	1.0 (±0.0)	0.2 (±0.0)	14.5
PTE	n.d.	n.d.	n.d.	13.4 (±0.8)	1.4 (±0.1)	32.6	n.d.	n.d.	n.d.
α-Car	0.1 (±0.0)	0.02 (±0.0)	n.d.	0.2 (±0.0)	1.8 (±0.1)	0.5	0.5 (±0.0)	0.05 (±0.0)	8.0
β-Car	11.5 (±0.7)	0.5 (±0.0)	10.2	11.2 (±0.6)	7.6 (±0.3)	27.3	3.3 (±0.1)	0.13 (±0.0)	49.2
Lyc	n.d.	n.d.	n.d.	4.2 (±0.2)	0.09 (±0.0)	10.2	0.01 (±0.0)	0.05 (±0.0)	0.2
Total car.	112.9 (±15.3)	1.3 (±0.1)	100	41.2 (±3.5)	0.7 (±0.0)	100	6.6 (±0.5)	0.2 (±0.0)	100

^1^ Absolute amount of carotenoid cellular uptake from Caco-2 cells following simulated GI digestion of each tested plant food matrices. ^2^ Cellular uptake represents the fraction of carotenoid recovered in the Caco-2 cell fraction compared to the original matrix content, considering all dilution factors. ^3^ Values represent the contribution (%) of each individual carotenoid, relative to the total amount of carotenoids taken up by the cells. Neo+Vio: neoxanthin+violaxanthin; Lut+Zea: lutein+zeaxanthin; PTF: phytofluene; PTE: phytoene; α-Car: α-carotene; β-Car: β-carotene; Lyc: lycopene. n.d.: not detectable.

## Data Availability

All relevant data are within the manuscript files.

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
