# Peer review of "Impact of Protein-Enriched Plant Food Items on the Bioaccessibility and Cellular Uptake of Carotenoids"

_antioxidants, 2021, doi:10.3390/antiox10071005_

Round 1

Reviewer 1 Report

This manuscript aims to study the effect of protein-enriched plant foods on the bioaccessibility and cellular uptake of carotenoids. In general this paper is well written and the technical quality is sound. However, the following issues should be addressed prior to possible publication of this paper:

  1. Both cod fillet and turkey fillet are used as protein source in this study. Will the fat content in cod fillet and turkey fillet affect carotenoids digestion and bioaccessibility?
  2. Will WPI, SPI, SC, GEL and proteins in cod fillet and turkey fillet be hydrolyzed in gastric fluid? If so, what is the degree of hydrolysis? What type of peptides can be formed? Will peptide affect carotenoid digestion and bioaccessibility?
  3. For carotenoid quantitation, both spectrophotometric and HPLC methods are used. It would be better to use only HPLC as the quantitative data can be more accurate.
  4. In Figure 1, a total of 6 protein sources are used. However, in Figure 3 and 4, only 4 protein sources are used. This point should be justified.
  5. Is it possible to study the effect of lutein or zeaxanthin on the bioaccessibility instead of a combination of both in the absence and presence of protein?
  6. The bioaccessibility order of individual carotenoid in tomato juice and carrot juice is different? What may cause this difference?
  7. The authors claimed that xanthophylls showed a better cellular uptake than carotenes. It would be better for the authors to provide the cellular uptake data of individual carotenoid standard and compare with that from spinach, carrot juice and tomato juice.

Author Response

Reviewer #1:

This manuscript aims to study the effect of protein-enriched plant foods on the bioaccessibility and cellular uptake of carotenoids. In general, this paper is well written and the technical quality is sound. However, the following issues should be addressed prior to possible publication of this paper:

Reply: We appreciate your kind assessment of the manuscript.

  1. Both cod fillet and turkey fillet are used as protein source in this study. Will the fat content in cod fillet and turkey fillet affect carotenoids digestion and bioaccessibility?

Reply: We believe that the small amount of fats could be considered negligible regarding the influence on the bioaccessibility of carotenoids, as cod and turkey fillets contain only 0.7 and 1.2% fat, respectively (ref. lines 156-164), and only 1.5 to 2.2 g of these proteins (equivalent to 50% RDA) were subjected to an in vitro digestion, so the total amount of fat coming from these sources would be low (<30 mg). In fact, cod and turkey were chosen as these were the lowest lipid-containing but protein-rich animal food matrices that we could think of.

  1. Will WPI, SPI, SC, GEL and proteins in cod fillet and turkey fillet be hydrolyzed in gastric fluid? If so, what is the degree of hydrolysis? What type of peptides can be formed? Will peptide affect carotenoid digestion and bioaccessibility?

Reply: We have evaluated the digestion of WPI, SPI, SC and GEL in our previous work, and polypeptides have been identified confirming the integrity and type of proteins 1. The influence of protein hydrolysis extent on carotenoids bioaccessibility has been also discussed in this earlier paper. Indeed, following gastric and intestinal digestions, hydrolysis appeared to be complete. The effects of turkey and cod proteins on carotenoid bioaccessibility were smaller than those of pure proteins, so that the latter were selected for the following experiments, while cod and turkey proteins were not retained for further analysis. Nevertheless, we understand the interest pointed out by the reviewer, and we have discussed proteolysis extent between the investigated proteins, including cod and turkey proteins, as reported by Montoya-Martinez et al 2 (lines 748-760).

  1. For carotenoid quantitation, both spectrophotometric and HPLC methods are used. It would be better to use only HPLC as the quantitative data can be more accurate.

Reply: We agree with the reviewer on the principle limitations of spectrophotometric methods. As mentioned in lines 237-241, this method was used only as a first evaluation of the impact of the proteins studied on total carotenoid micellization. We have shown in our previous work that the spectrophotometric methods based on the mean absorption wavelength and coefficient of the most abundant carotenoids in plants resulted in a very close estimation of the carotenoid content compared to HPLC, including the matrices carrots, tomato and spinach 3. For this present study, we have now provided the measurements of the total carotenoids from extracts of the investigated matrix extracts; carried out both by HPLC and by spectrophotometer. The results attached below show that the latter method can be seen as equal to results obtained by HPCL (please refer to figure below).

We would also like to point out that if any inaccuracy should have occurred, leading to a slight over-or underestimation of total carotenoid content, this should not have affected the general results obtained in the present study, given that all the different conditions (type of protein and concentrations) were compared to the same control.

  1. In Figure 1, a total of 6 protein sources are used. However, in Figure 3 and 4, only 4 protein sources are used. This point should be justified.

Reply: As mentioned above, turkey and cod proteins had a less strong impact compared to pure proteins on the bioaccessibility of total carotenoids. Therefore, we decided to continue studying in more detail the effect of these proteins on the bioaccessibility of individual carotenoids.

  1. Is it possible to study the effect of lutein or zeaxanthin on the bioaccessibility instead of a combination of both in the absence and presence of protein?

Reply: We have chosen to combine the two carotenoids in order to avoid possible errors in quantification, due to the two slightly overlapping spectra of lutein and zeaxanthin. It should be noted that the majority was lutein in all matrices and only a small amount of zeaxanthin was detected, in line with literature data tha zeaxanthin levels are rather low in the employed food matrices spinach. Regrettably, this is not an uncommon limitation to the HPLC-based quantification of carotenoids.

  1. The bioaccessibility order of individual carotenoid in tomato juice and carrot juice is different? What may cause this difference?

Reply: We think that matrix effect could be one of the reasons why tomato juice exhibited a higher total bioaccessibility than carrot juice, e.g. thickness and rigidness of the cell walls etc., please refer to lines 677-681 for more details.

  1. The authors claimed that xanthophylls showed a better cellular uptake than carotenes. It would be better for the authors to provide the cellular uptake data of individual carotenoid standard and compare with that from spinach, carrot juice and tomato juice.

Reply: Our findings were supported by previous studies reporting the same results (lines 763-768). For cellular uptake experiments, samples obtained from in vitro digestion were used in order to ensure the same experimental parameters for each condition (type of protein and concentration, type of matrices…etc.). We believe that the use of pure standards alone could lead to different results, due to changes such as matrix effects, as such, standards would require artificial solubilization/micellization at digestion onset. Hence, we found it more realistic to compare the cellular uptake from realistic food matrices against one another.

Reviewer 2 Report

In this paper by Iddir et al., the Authors investigated the effect of different type (SPI, SC, GEL, turkey and cod ) and concentration (from 10 to 50 % of RDA) of protein rich foods  on carotenoids bioaccessibility and bioavailability in caco-2 cells. The results have shown as the type of proteins and their concentration may differently influence carotenoids release in the micellar fraction during in vitro digestion and their subsequent uptake in intestinal cells.  The Authors conclude that carotenoid bioaccessibility may be strongly influenced by the cellular localization and by the hydrophobicity of carotenoids and proteins which may interact with the interfaces of micelles. The study is elegant, well written, the methodologies used are pertinent and the discussion are supported by the results and by the scientific literature.

I have only minor considerations:

  1. Line 187-190. The procedure to obtain the micellar fraction is essential in this study and the Authors should explain it in detail instead cite the reference.
  2. line 202-203. Normally, caco-2 cells take 21 days to be completely differentiated and membrane integrity is usually evaluated measuring TEER (i.e.: 10.1016/j.foodres.2019.108940). The Authors should clarify this point.
  3. Line 205. Is the complete digesta filtered on 0.22um filters before to be supplemented to cells? This point is crucial cause filtration may remove lipid micelles. Please specify.
  4. In preliminary experiments, cod and turkey have been chosen as protein rich food but in subsequent experiment only WPI, SPI, SC and GEL were used. If they are skipped to their low protein digestibility or other reasons this should be indicated in the text.
  5. Provide a higher quality figures (particularly for the figure 6C that seems a screenshot ).
  6. In figure 6 the statistical analysis is missing.
  7. paragraph 3.7.1. Can you indicate in the text the average size of lipid droplet?
  8. Line 362. Figure 1 legend. The first part include the formatting guide.
  9. Some descriptions of results in the text seem not correspond to the figure.

Line 419. The presence of proteins had not a negative effect on micellization of lut+zea in all matrices, in fact SC 50 improve it, while WPI and GEL had no effect.

Line 444-445. The Authors state that the presence of SPI resulted in highest beta-carotene bioaccessibility, followed by SC, WPI and GEL. But this is not completely true cause SC and GEL reduce it when came from spinach and tomato juice.

Line 541-543. Similarly as reported above.

Line 564-567. This is true only in carrot juice but not for spinach.

Line 575-576. Not in all tested concentration but only in 25 and 50. Statistical symbols in WPI 10 is not present.

Please check carefully the whole manuscript.

Author Response

Reviewer #2:

In this paper by Iddir et al., the Authors investigated the effect of different type (SPI, SC, GEL, turkey and cod) and concentration (from 10 to 50 % of RDA) of protein rich foods on carotenoids bioaccessibility and bioavailability in caco-2 cells. The results have shown as the type of proteins and their concentration may differently influence carotenoids release in the micellar fraction during in vitro digestion and their subsequent uptake in intestinal cells.  The Authors conclude that carotenoid bioaccessibility may be strongly influenced by the cellular localization and by the hydrophobicity of carotenoids and proteins which may interact with the interfaces of micelles. The study is elegant, well written, the methodologies used are pertinent and the discussion are supported by the results and by the scientific literature.

Reply: We appreciate the overall assessment and remarks of the referee.

I have only minor considerations:

  1. Line 187-190. The procedure to obtain the micellar fraction is essential in this study and the Authors should explain it in detail instead cite the reference.

Reply: We thank the reviewer for this comment, more details are now added (lines 187 – 189).

  1. line 202-203. Normally, caco-2 cells take 21 days to be completely differentiated and membrane integrity is usually evaluated measuring TEER (i.e.: 10.1016/j.foodres.2019.108940). The Authors should clarify this point.

Reply: We agree with the reviewer that a period of 3 weeks has been recommended by some authors. However, based on our previous studies 4, 5, maintaining cell culture for 2 weeks has been sufficient to allow for a differentiation of Caco-2 cells based on optical investigation and TEER measures. It could be argued that certain expression of transporters may still change with prolonged periods of time, but then again also the Caco-2 cell model has its limitations, possibly also depending on the passages and strain used. However, we have added a remark in the materials and methods, please see lines 205-206.

  1. Line 205. Is the complete digesta filtered on 0.22um filters before to be supplemented to cells? This point is crucial cause filtration may remove lipid micelles. Please specify.

Reply: We thank the reviewer for pointing this out, it is now specified (see line 209). No filtration was done in this case, as this would be non-physiological, i.e. all digesta would be passed on from the stomach to the small intestine eventually.

  1. In preliminary experiments, cod and turkey have been chosen as protein rich food but in subsequent experiment only WPI, SPI, SC and GEL were used. If they are skipped to their low protein digestibility or other reasons this should be indicated in the text.

Reply: We appreciate the comment and we have followed the reviewer’s recommendation (see lines 410-413). Cod and turkey were not further investigated as their influence on carotenoid bioaccessibility appeared smaller compared to other proteins.

  1. Provide a higher quality figures (particularly for the figure 6C that seems a screenshot ).

Reply: The figure 6C is now changed.

  1. In figure 6 the statistical analysis is missing.

Reply: We agree with the reviewer, but the superscripts to show the significant differences have been intentionally omitted in this figure, which was intended to only give an idea of the overall effect of the co-digested proteins being low and limited. In fact, most conditions differ significantly from one another, though the physiological effects (viscosity changes <ca. 25%) were not assumed to be physiolgically relevant.  However, we have added the statistical interpretation now to the figure.

  1. paragraph 3.7.1. Can you indicate in the text the average size of lipid droplet?

Reply: We apologize; we did not specify in the figure 5 legend what the white bar refers to, which could surely meet the reviewer's request. It is now added (see line 624).

  1. Line 362. Figure 1 legend. The first part include the formatting guide.

Reply: We thank the referee; the formatting guide is now removed.

  1. Some descriptions of results in the text seem not correspond to the figure. In line 419, the presence of proteins had not a negative effect on micellization of lut+zea in all matrices, in fact SC 50 improve it, while WPI and GEL had no effect.

Reply: In line 419, we have reported the statistical results on the overall effect of each protein type (all concentrations and matrices combined) following linear mixed model (please see line 426). The effect of protein concentrations was reported in lines 431 to 442. We have tried to further clarify this within the text now.   

  1. Line 444-445. The Authors state that the presence of SPI resulted in highest beta-carotene bioaccessibility, followed by SC, WPI and GEL. But this is not completely true cause SC and GEL reduce it when came from spinach and tomato juice.

Reply: We agree with the reviewer regarding the effects of SPI in spinach and tomato juice, but in line 449, we again, reported the overall effect of SPI in all matrices and concentrations combined. We are aware that this is perhaps slightly confusing and have tried to emphasize this better in the statistical section and the results now.

  1. Line 541-543. Similarly as reported above.

Reply: We performed the same statistical analysis as for the bioaccessibility; we have tried to be more precise in the text now.

  1. Line 564-567. This is true only in carrot juice but not for spinach.

Reply: We thank the referee for this comment; it is now corrected (line 579).

  1. Line 575-576. Not in all tested concentration but only in 25 and 50. Statistical symbols in WPI 10 is not present.

Reply: We agree with the reviewer, not in all tested concentrations, but again, here we reported the overall effect of proteins (concentrations and matrices combined/pooled). Again, we tried to clarify this in the text.

  1. Statistical symbols in WPI 10 is not present.

Reply: We thank the referee, it is now added.

  1. Please check carefully the whole manuscript.

Reply: We followed the reviewer’s recommendation, and we have paid attention to the descriptions of results and tried to precise interpretation of results, especially when talking about pooled/combined findings of several conditions.

Reviewer 3 Report

The manuscript entitled “Impact of protein-enriched plant Food items on the bioaccessibility and cellular uptake of carotenoids” evaluate the effect of diverse proteins on the bioaccessibility and cellular uptake of carotenoids from selected food sources. The manuscript is well organized, and discussed. This manuscript should be accepted in Antioxidants after minor revisions.

Table 1. Please add the standard deviation

The figure resolution should be improved.

Author Response

Reviewer #3:

The manuscript entitled “Impact of protein-enriched plant Food items on the bioaccessibility and cellular uptake of carotenoids” evaluate the effect of diverse proteins on the bioaccessibility and cellular uptake of carotenoids from selected food sources. The manuscript is well organized, and discussed. This manuscript should be accepted in Antioxidants after minor revisions.

Reply: We appreciate the overall assessment and remarks of the referee.

  1. Table 1. Please add the standard deviation.

Reply: It is now added.

  1. The figure resolution should be improved.

Reply: The referee’s suggestion was followed. It is also possible that during the transformation into the PDF file the resolution was decreased compared to the word-file.
